# The CSHQ-DE Questionnaire Uncovers Relevant Sleep Disorders in Children and Adolescents with Long COVID

**DOI:** 10.3390/children9091419

**Published:** 2022-09-19

**Authors:** Sarah Werner, Claus Doerfel, Richard Biedermann, Michael Lorenz, Marius Rasche, Hans Proquitté, Lars Newman, Daniel Vilser

**Affiliations:** 1Neonatal and Pediatric Intensive Care Section, Department of Child and Adolescent Medicine, University Hospital Jena, 07747 Jena, Germany; 2Respiratory, Allergy and Cystic Fibrosis Section, Department of Child and Adolescent Medicine, University Hospital Jena, 07747 Jena, Germany; 3Cardiology Section, Department of Child and Adolescent Medicine, University Hospital Jena, 07747 Jena, Germany

**Keywords:** long COVID, post-COVID-19 condition, sleep disorders, SARS-CoV-2, coronavirus, children

## Abstract

Acute SARS-CoV-2 infections in children and adolescents are usually mild. However, they can suffer from ongoing symptoms, generally referred to as long COVID. Sleep disorders are one of the most frequent complaints in long COVID although precise data are missing. We assessed the sleep behavior of children and adolescents who presented at our outpatient clinic between January 2021 and May 2022 with the Children’s Sleep Habits Questionnaire (CSHQ-DE). We compared the sleep behavior at three different time points: pre-COVID-19; post-COVID-19 at the initial presentation; and post-COVID-19 at re-presentation. Data from 45 patients were analyzed. Of those, 64% were female and the median age was 10 years (range: 0–18 years). Asymptomatic or mild COVID-19 disease was experienced in 89% of patients; 11% experienced moderate disease. The initial presentation occurred at a median of 20.4 weeks (6 weeks–14 months) after the infection. The CSHQ-DE score increased significantly from pre-COVID-19 (45.82 ± 8.7 points) to post-COVID-19 (49.40 ± 8.3 points; *p* ≤ 0.01). The score then normalized at re-presentation (46.98 ± 7.8; *p* = 0.1). The greatest changes were seen in the CSHQ-DE subscale score “daytime sleepiness”. Our data showed that children and adolescents with long COVID often suffer from sleep disturbances. For most children and adolescents, these sleep disorders decreased over time without any further medical intervention aside from a basic sleep consultation.

## 1. Introduction

Acute SARS-CoV-2 infections in children and adolescents are usually mild [1]. Similar to adults, children can suffer from ongoing symptoms [2]. This new disease is generally referred to as long COVID, post-COVID-19 condition or post-COVID-19 syndrome [3]. NICE defines long COVID in children, adolescents and adults as persisting symptoms that develop during or after a COVID-19 infection and are present for more than 4 weeks (ongoing COVID-19) or 12 weeks (post-COVID-19 syndrome) [4]. A slightly different definition exists from the WHO, which currently only refers to adults [5]. Worsening symptoms in a pre-existing disease as a result of a SARS-CoV-2 infection are also assigned to long COVID, according to the German long COVID guideline [6]. Over 200 symptoms are described and any organ may be affected [7,8,9,10]. The most frequently described complaints are fatigue, respiratory problems, cognitive dysfunction, cardiac symptoms, abdominal pain and sleep disturbances [11,12,13]. Whilst it is known that sleep disorders have a significant impact on the quality of life of children and adolescents [14], precise data, especially longitudinal data relating to the effect of long COVID on sleep in children and adolescents, are lacking. The adequate care of sick children and adolescents requires a holistic approach in order to address all aspects of the illness. Therefore, we routinely assessed the sleep behavior of children and adolescents who presented to our outpatient long COVID clinic.

## 2. Materials and Methods

### 2.1. Cohort Composition

From January 2021 to May 2022, children and adolescents who presented to the interdisciplinary long COVID outpatient clinic of the Children’s Hospital of Jena, Germany, were screened for sleeping disorders using the CSHQ-DE. Patients who fulfilled the long COVID diagnostic criteria and consented to the study were included in the study. A total of 102 children and adolescents met this criteria, of which 45 completed the questionnaires in full at all three time points and were included in the analysis. The most common reason for exclusion was that the questionnaire was not completed at the time of re-presentation as the symptoms had resolved.

### 2.2. CSHQ-DE

The sleep-related symptoms of patients were assessed using the standardized Children’s Sleep Habits Questionnaire (CSHQ-DE) for clinically relevant sleeping issues of school-aged children (4–10 years) [15,16,17]. In this study, the CSHQ-DE was also used to screen sleep disorders of children and adolescents outside of this age range (0–18 years). The questionnaire provides a total score and eight subscale scores, which reflect the key sleep domains. The clinical cut-off used for a possible sleep dysfunction was a total score ≥ 41 points. The CSHQ-DE was usually parent-reported, with adolescent patients independently filling out the questionnaire if able to do so. The CSHQ-DE was completed at three time points: pre-COVID-19 (retrospectively); post-COVID-19 at the initial presentation; and post-COVID-19 at re-presentation (three months later). Only patients for whom the CSHQ was completed in its entirety at all three time points and whose complaints met the diagnostic criteria for long COVID were included in the analysis. All parents gave their written informed consent to the participation of their children in the study. The study protocol was retrospectively approved by the local Ethics Committee (Protocol No. 2022-2614_1-BO).

### 2.3. Further Investigations

A diagnosis of long COVID necessitates that alternative diagnoses are excluded. Additionally, residual organ damage following a SARS-CoV-2 infection should be detected. Therefore, all children presenting to our long COVID clinic underwent extensive investigations. In addition to a detailed history and physical examination, all patients were investigated with an echocardiography, ECG, vital signs (including blood pressure) and extensive blood and urine tests. Patients who were able to cooperate underwent an orthostatic test, an exercise test, a pulmonary function test, an endothelial dysfunction test (dynamic vessel analysis) and a psychosocial review. If indicated, the assessment was smented with additional investigations, including sonography, MRI/CT (lung, brain, abdomen, renal and joints), EEG, a hand strength measurement and stool pathology.

### 2.4. Statistical Analysis

Data were analyzed using SPSS 28. As most of the data were normally distributed, the data were presented as the mean +/− SD (standard deviation). The groups were analyzed using a univariate ANOVA with repeated measures for multiple time points. Multiple measurements were corrected using a Bonferroni post hoc analysis. Multivariate regressions between the sleep quality indices and the symptoms of long COVID were performed. A *p*-value < 0.05 was considered to indicate a statistical significance (two-tailed).

## 3. Results

Data from 45/102 (44.1%) patients presenting for the first time at the long COVID outpatient clinic between January 2021 and May 2022 were analyzed. Of those, 64% were female and the median age was 10 years (range: 0 to 18 years). Asymptomatic or mild COVID-19 disease was experienced in 89% of patients; 11% experienced moderate disease. The average body mass index (BMI) mean percentile, based on CDC values, was 74% (IQR 39, 89).

Fifteen (33%) patients had a current or previous medical issue (Table 1). Six children had more than one current or previous medical issue. The most common disease group was pulmonary/allergic disease (10/45), with 7 patients (16%) having a history of atopy (asthma/eczema/allergies). Two patients (who were siblings) had cystic fibrosis, two patients had obesity and two patients had a current seizure disorder. One patient with a diagnoses of metabolic syndrome and Rolandic epilepsy was medicated with gabapentin, enalapril and metformin. One patient had absence seizures on no medications and a further patient had an autoimmune hepatitis/primary sclerosing hepatitis overlap and was managed with prednisone and azathioprine. Of note, no patients had a diagnosis of obstructive sleep apnea.

The initial presentation occurred at a median of 20.4 weeks (6 weeks to 14 months) after the infection. The assessment was repeated at the scheduled re-presentation 3 months later (mean 14.3 weeks; range 7 weeks to 9 months). At the initial presentation, 80% of the patients suffered from two or more symptoms (Table 2). The most common symptoms were fatigue/stress intolerance (62%), concentration/learning difficulties (44%), sleep disorders (42%), pulmonary symptoms (38%) and headaches (36%). The differentiation of symptoms by age is shown in Table 3.

From their history, 42% of patients reported sleep disorders post-COVID-19. At the initial presentation, the CSHQ-DE showed that 88.9% suffered from a possible sleep dysfunction (total score ≥ 41); however, 66.7% already had a possible sleep dysfunction pre-COVID-19. Of the patients who self-reported suffering from sleep disorders post-COVID-19, 94.7% had a total CSHQ-DE score ≥ 41. Pre-COVID-19, the mean CSHQ-DE score of all 45 assessed patients was 45.82 ± 8.7 points, increasing significantly to 49.40 ± 8.3 points (*p* ≤ 0.01) post-COVID-19. Thereafter, a trend to normalize (46.98 ± 7.8; *p* = 0.1) was observed at re-presentation (Figure 1). In an age-group subanalysis (Table 4), a statistical difference was shown for the CSHQ scores of the 6–11 (*p* = 0.035) and 12–18 (*p* = 0.007) years age groups. No statistical difference was found for the 0–5 year age group.

Of note, the CSHQ-DE subscale score “daytime sleepiness” significantly increased from pre- to post-COVID-19 (13 ± 3.0 vs. 14 ± 3.2, respectively; *p* ≤ 0.001) and significantly decreased again at the follow-up (13 ± 3.2; *p* = 0.029) (Figure 2). All other subscale scores of the CSHQ-DE did not display a change from pre- to post-COVID-19 (Table 5).

The multivariate regression analysis showed no significant effect of the occurrence of fatigue/stress intolerance or neurological, cardiac or pulmonary long COVID symptoms on either the CSHQ-DE score or daytime sleepiness. The presence of a pre-existing medical condition did not affect the risk of developing a sleep disorder after a SARS-CoV-2 infection compared with previously healthy individuals (data not shown). Individual comorbidities could not be meaningfully statistically analyzed due to the small sample size.

## 4. Discussion

The presented data showed that sleep disturbances—most notably, an increase in daytime sleepiness—occurred in children and adolescents with long COVID. As has already been described in adults [18], a sleep dysfunction is independent of the disease severity. The distribution of the symptoms was consistent with other studies of long COVID in children and adolescents, including a female preponderance [3,11,19,20]. The patients underwent a thorough history and diagnostics with the aim of identifying alternative diagnoses that may have been responsible for the presenting symptoms. As such, we were confident that the children and adolescents included in this study had long COVID. The study patients had no diagnoses of obstructive sleep apnea, narcolepsy or hypertension (aside from a patient with previously diagnosed metabolic syndrome). Two patients were taking medication during the study period that, theoretically, may have had an influence on the sleep behavior. One patient was on regular gabapentin due to epilepsy, for which the dose remained unchanged from prior to the infection and throughout both assessments. In our opinion, gabapentin had no influence on the sleep behavior of that patient. The second patient, who had liver disease, commenced with prednisolone between the initial presentation and re-presentation. Their sleep behavior was markedly improved at re-presentation; however, an effect secondary to prednisolone could not be excluded. For all other patients, there were no medications considered causative for sleep disorders. Our results showed a correlation between long COVID and changes in sleep behavior and we assumed the disease to be causal. Our study did not provide a pathophysiological explanation for this relationship.

Various pathophysiological causes of sleep disturbances secondary to a SARS-CoV-2 infection are discussed in the literature [21,22]. The mechanisms include long-term tissue damage, viral persistence and chronic inflammation. The production of pro-inflammatory cytokines in the central nervous system—in particular, the brainstem—can cause post-viral symptoms [23]. Several cytokines and viral envelope proteins are hypnogenic and have specific effects on both NREM and REM sleep. Certain phases of the infection and the strength of the immune response may be arousal-inducing and cause sleep disruption [24,25]. A long-term consequence is disturbed sleep–wake cycles [26]. Moreover, it appears that the systemic inflammation triggered by the virus and consecutive activation of inflammatory mediators such as interferons and interleukin-6 can impair neuronal functions and cause CNS symptoms [27]. Inflammatory mediators may, in turn, contribute to the manifestation of autoimmune processes [28]. There is evidence for an underlying autoimmune mechanism, as shown by persistently elevated antinuclear antibody (ANA) [29] and G-protein-coupled receptor antibody titers [30] in long COVID patients. In this regard, a typically high female proportion in long COVID cohorts after mild disease is consistent with the known female predominance in autoimmunity [31].

Differentiating sleep disturbances, especially daytime sleepiness, from fatigue is difficult. The pathogenesis of fatigue after COVID-19 remains unclear and is currently under investigation. COVID-19-related organ damage as well as psychological disturbances can be related to the development of fatigue [32]. Persisting inflammation or an autoimmune response due to antigen mimicry may also play a role [33]. In addition, non-restorative sleep is associated with daytime sleepiness, which can contribute to fatigue and cognitive dysfunctions. Therefore, sleep disturbance in long COVID overlaps with, and is difficult of differentiate from, fatigue and cognitive dysfunctions. This highlights the importance of a differentiated assessment of sleep disturbances vs. fatigue with the aim of guiding treatment [34].

Furthermore, it remains difficult to separate the infectious sequelae of SARS-CoV-2 from factors relating to the broader pandemic (e.g., lockdowns and school closures). There is a paucity of data relating to the effect of lockdowns and school closures on screen time, social interaction and exercise, which are known risk factors for poor sleep quality. The effect of the broader pandemic, coupled with changes in electronic device and social media behavior, can induce stress, anxiety and weight gain. These, in turn, exacerbate sleep dysfunction [35].

Two-thirds of our patients already had an abnormal CSHQ-DE score pre-COVID-19 infections. In adult patients, there is evidence that poor sleep is associated with a higher susceptibility to a SARS-CoV-2 infection as well as worse clinical outcomes [36,37,38]. It is possible that poor sleep results in the delayed resolution of symptoms; however, evidence for this in children is lacking.

The impact of chronically disturbed sleep on the development of long COVID remains unknown. A sleep deficiency can contribute to cell stress and consecutive neuronal damage [39]. In addition, sleep disorders can induce mild inflammation, causing blood–brain barrier dysfunctions and allowing antigens and inflammatory mediators passage into the CNS. In this respect, pre-existing sleep disturbances in the context of long COVID could adversely influence the long-term course of COVID-19 disease [34].

Each patient underwent a thorough assessment of their past medical history; 15/45 (33%) had a current or resolved medical condition. Once patients with resolved/clinically non-significant diseases were excluded (marked with * in Table 1), 12/45 (26.7%) patients were identified as having an ongoing medical issue. This value was slightly higher than the KIGGS data, which displayed that the prevalence of chronic disease in 0 to 17-year-old patients in Germany was 16.2% [40]. Our findings were consistent with recently published data from the USA [41], which showed that children with pre-existing medical conditions had a higher risk of developing long COVID. Our cohort was, however, too small to reliably assess the effects.

As discussed above, it remains difficult to differentiate sleep disorders from daytime fatigue. Regardless of the etiology, our study showed that this was a relevant problem for children and adolescents following a SARS-CoV-2 infection. Only 42% of the patients reported sleep difficulties in the medical history at the initial presentation; the CSHQ-DE showed that 89% of patients had a sleep dysfunction. Given that there is currently no causal therapy for the treatment of long COVID, symptomatic treatment options must be exhausted for all sufferers.

### Treatment Recommendations

Our approach to the treatment of sleep disorders was analogous to the recommended treatment of non-organic sleep disorders by the German guideline [42]. The most important intervention was detailed advice on sleep hygiene (Table 6). A sleep diary with every attempt at sleep modifications was encouraged in order to monitor the effectiveness of the intervention.

In cases of severe sleep disorders, pharmacotherapy should be considered. The choice of medication should depend on the type of sleep disorder and the underlying disease (e.g., depression or panic disorders) [42]. The dose will vary greatly between patients and should, therefore, be carefully implemented. Most of the medications available are prescribed off-label in Germany and can, therefore, only be used as part of an individual treatment regime.

Throughout our study, we initiated pharmacotherapy in only four patients. Melatonin was used in increasing doses in three adolescents over a maximum period of six weeks. A 15-year-old boy and 13-year-old girl with issues relating to sleep initiation, sleep maintenance and daytime tiredness experienced a significant improvement after 4 and 6 weeks, respectively. This improvement persisted even after discontinuation. Additionally, a 17-year-old girl with severe sleep initiation issues was treated with melatonin resulting in no improvement. A fourth child (a 9-year-old girl) had only mild problems with sleep initiation and was treated with dimetindene (Fenistil) primarily due to recurrent urticaria. Both symptoms improved with the medication.

Our first choice for the treatment of moderate and severe sleep disorders secondary to long COVID was melatonin. Taken regularly for a few weeks, melatonin restores the regular circadian rhythm [43]. It has antioxidant, anti-inflammatory and immunomodulatory properties [44,45] and thus can have a positive effect on the previously discussed mechanisms of long COVID [7,9,46]. Melatonin accumulates in the mitochondria of endothelial cells, resulting in beneficial effects on sepsis-induced mitochondrial dysfunctions, oxidative stress and the cytokine response [47].

Endothelial dysfunctions occur in acute COVID-19 infections [48,49,50,51,52,53]. Endothelial dysfunctions caused by persistent endotheliitis could explain many of the symptoms of long COVID [54,55,56,57,58]. Melatonin activates nuclear factor erythroid-derived 2-like 2 (NRF2) [59]. NRF2 is the primary regulator of the inducible antioxidant response, which attenuates the cellular injury in oxidative stress [59]. Melatonin has been explored as a treatment for other viral infections that produce an excessive inflammatory response [60,61,62]. Its use in COVID-19 has also been proposed [63] and is part of several international trials [64].

We were restrictive in the treatment of sleep disorders with medications. Given the potential of melatonin to have a positive effect on many aspects of the discussed pathogeneses of long COVID coupled with minimal side effects, we are currently considering an earlier use. Future studies should focus on the effect of melatonin on clusters of long COVID symptoms and its use independently of sleep disorders.

## 5. Strengths and Limitations

A strength of our study was the detailed assessment of symptoms, which were obtained via a physician-led interview and questionnaires. The temporal relationship between a SARS-CoV-2 infection and the symptoms was assessed in detail. Extensive diagnostics assisted in exclusion of alternate diagnoses as the cause of the symptoms. Therefore, the diagnosis of long COVID was made as accurately as possible, according to the available criteria.

One limitation of this study was the absence of a control group. Further studies on this topic would benefit from a control to correct for confounders of the broader pandemic and non-SARS-CoV-2 viral illnesses. It was demonstrated in adults that sleep disorders were more prevalent among patients who had recovered from COVID-19 than the non-infected control group [65]. A systematic review from 2022 of the effects of coronavirus disease on sleep in children aged 12 years and younger revealed an increase in the sleep duration, sleep latency and daytime sleepiness. Reliable data regarding the long-term effects on the sleep and daytime function of children are lacking [66]. Another limitation was that the pre-COVID sleep assessment was retrospectively conducted, which may have resulted in recall bias. Due to the nature of the study, a concurrent assessment was not possible.

Our results were limited by the fact that a large proportion of patients who presented to our long COVID outpatient clinic were not included in the study. The study only included patients who fulfilled the diagnostic criteria for long COVID; these are highly subjective and lacking in objective measurements such as biomarkers. We used the same diagnostic criteria that were later published in the “*Recommendation for standardized medical care for children and adolescents with long COVID*” of the German Society for Pediatrics (DGKJ) [12]. Of the 102 who presented to the clinic, only 45 families completed the questionnaire in full at all three time points. The two primary reasons for non-completion were the omission of answers to individual questions or, above all, the symptoms had resolved and the parents, therefore, considered the completion of the questionnaire to be unnecessary. We believe that the inclusion of the missing questionnaires would have further highlighted the group differences.

Although patients were questioned regarding pre-existing medical conditions, they were not routinely screened for all possible organic causes of a sleep disorder. Polysomnography was only performed when the history suggested a central or obstructive respiratory disorder. This was not the case for any of the patients included in this study.

## 6. Conclusions

Our results suggested that sleep disturbances are common in children with long COVID. Of the children and adolescents in our study, 89% showed abnormalities in sleep behavior when using an objective questionnaire. The increase in the CSHQ-DE score from 45.8 to 49.4 due to the infection was significant. Therefore, the monitoring of sleep behavior should occur in all patients with long COVID. If the sleep disturbance does not improve with counselling, medication should be used. In our opinion, melatonin is the first-line medication, due to the improvements in sleep, the good side-effect profile and the potential beneficial effects on the underlying long COVID pathophysiology.

## Figures and Tables

**Figure 1 children-09-01419-f001:**
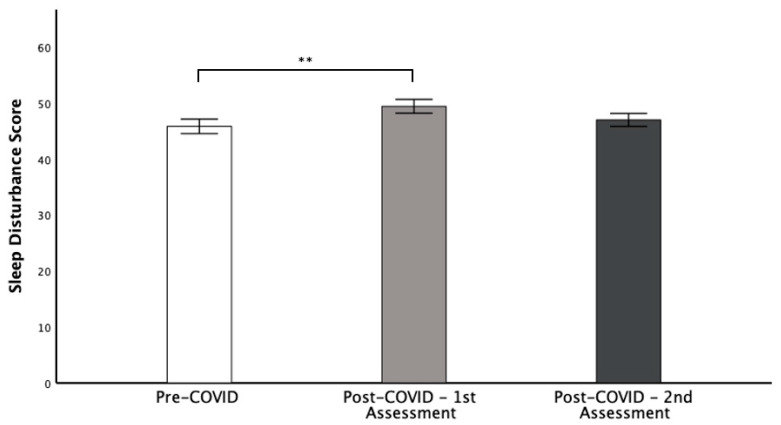
Total CSHQ-DE score pre-COVID-19 vs. post-COVID-19 initial and re-presentation. The sleep disturbance score significantly increased between pre-COVID-19 and 1st assessment (45.82 ± 8.7 to 49.40 ± 8.3, respectively), then decreased (46.98 ± 7.8). Univariate ANOVA with multiple measurements, post hoc Bonferroni (** *p* ≤ 0.01).

**Figure 2 children-09-01419-f002:**
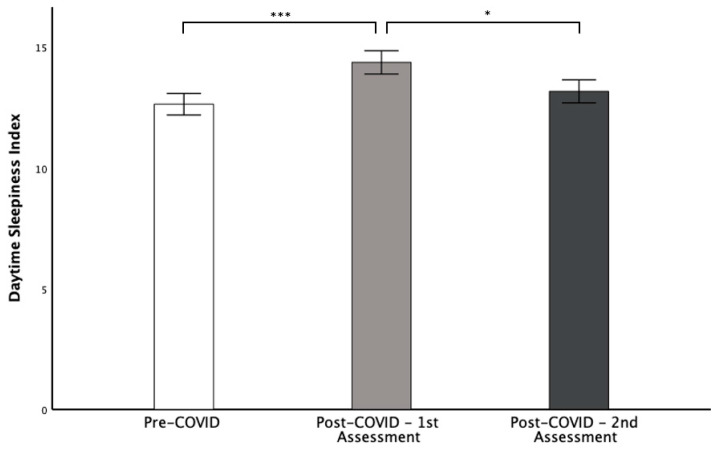
Subscale score “Daytime Sleepiness” pre-COVID-19 vs. post-COVID-19 initial and re-presentation: Significant increase from pre- to post-COVID-19 (13 ± 3.0 vs. 14 ± 3.2, respectively) and significant decrease at re-presentation (13 ± 3.2). Univariate ANOVA with multiple measurements and post hoc Bonferroni (*** *p ≤* 0.001 and * *p* = 0.029, respectively).

**Table 1 children-09-01419-t001:** Patient characteristics and medical history.

Characteristics	*n*	Distribution
*n*	45	
Age, mean in years	45	10.0 (IQR 8.0, 14.0)
Female gender	45	29 (64%)
BMI percentile, mean (based on CDC)	45	74.0 (IQR 39.0, 89.0)
COVID-19 Course	45	
-Asymptomatic		5 (11%)
-Mild		35 (78%)
-Moderate		5 (11%)
Medical History		
Pulmonary/allergic diseases - Cystic fibrosis - Asthma/reactive airways - Other allergy/eczema - RDS, pneumothorax or allergy - Recurrent croup *		10 (22%) −2 −3 −3 −1 −1
Neurological diseases - Seizure disorder - Migraine - Developmental delay - Febrile seizures *		5 (11%) −2 −1 −1 −1
von Willebrand disease (type 1)		1
Right-sided aortic arch * Autoimmune hepatitis or primary sclerosing cholangitis Constipation		1 1 1
Obesity Metabolic syndrome		2 1
Psychogenic gait disorder or somatization disorder		1
Williamsbeuren Syndrome		1
Prematurity (32 and 34 weeks gestation)		2

* indicates disease that was either resolved at the time of infection or was considered to be a normal variant.

**Table 2 children-09-01419-t002:** Predominant symptoms at the initial presentation.

Symptoms	In Total (%)
Fatigue/stress intolerance	30/45 (62%)
Concentration/learning difficulties	20/45 (44%)
Sleep disorders	19/45 (42%)
Pulmonary symptoms	17/45 (38%)
Headache	16/45 (36%)
Gastrointestinal symptoms	14/45 (31%)
Cardiac symptoms	10/45 (22%)
Dizziness	10/45 (22%)
Muscle and joint symptoms	9/45 (20%)
Neurological symptoms	9/45 (20%)
Smell and taste disorders	9/45 (20%)
Hair loss	6/45 (13%)
Skin changes	4/45 (9%)

**Table 3 children-09-01419-t003:** Predominant symptoms at initial presentation; differentiation by age.

Symptoms	In Total (%)
Fatigue/stress intolerance	30/45 (62%)
0–5 years	5/9 (56%)
6–11 years	11/18 (61%)
12–18 years	12/18 (67%)
Concentration/learning difficulties	20/45 (44%)
0–5 years	1/9 (11%)
6–11 years	10/18 (56%)
12–18 years	9/18 (50%)
Sleep disorders (reported in the medical history)	19/45 (42%)
0–5 years	3/9 (33%)
6–11 years	7/18 (39%)
12–18 years	9/18 (50%)
Pulmonary symptoms	17/45 (38%)
0–5 years	5/9 (55.6%)
6–11 years	5/18 (28%)
12–18 years	7/18 (39%)
Headache	16/45 (36%)
0–5 years	0/9 (0%)
6–11 years	8/18 (44%)
12–18 years	8/18 (44%)
Gastrointestinal symptoms	14/45 (31%)
0–5 years	4/9 (44%)
6–11 years	3/18 (17%)
12–18 years	7/18 (39%)
Cardiac symptoms	10/45 (22%)
0–5 years	0/9 (0%)
6–11 years	4/18 (22%)
12–18 years	6/18 (33%)
Dizziness	10/45 (22%)
0–5 years	2/9 (22%)
6–11 years	2/18 (11%)
12–18 years	6/18 (33%)
Muscle and joint symptoms	9/45 (20%)
0–5 years	2/9 (22%)
6–11 years	3/18 (17%)
12–18 years	4/18 (22%)
Neurological symptoms	9/45 (20%)
0–5 years	2/9 (22%)
6–11 years	4/18 (22%)
12–18 years	3/18 (17%)
Smell and taste disorders	9/45 (20%)
0–5 years	0/9 (0%)
6–11 years	2/18 (11%)
12–18 years	7/18 (39%)
Hair loss	6/45 (13%)
0–5 years	1/9 (11%)
6–11 years	1/18 (6%)
12–18 years	4/18 (22%)
Skin changes	4/45 (9%)
0–5 years	1/9 (11%)
6–11 years	3/18 (17%)
12–18 years	0/18 (0%)

**Table 4 children-09-01419-t004:** Subgroup analysis of the CSHQ-DE score differentiated by age group (two-sided ANOVA (Friedman)).

Age Group	Total CSHQ-DE			*p*-Value
	Pre-COVID	Post-COVID 1st Assessment	Post-COVID 2nd Assessment	
0–5 years	49.0 (IQR 46.0, 61.0)	55.0 (IQR 53.0, 58.0)	48.0 (IQR 45.0, 53.0)	0.131
6–11 years	46.5 (IQR 40.0, 52.25)	54.0 (IQR 45.0, 56.0)	47.0 (IQR 44.0, 55.75)	0.035
12–18 years	43.0 (IQR 38.25, 44.0)	45.0 (IQR 41.0, 46.75)	43.5(IQR 40.25, 46.75)	0.007

**Table 5 children-09-01419-t005:** CSHQ-DE subscales, mean (SD), ANOVA with multiple measurements and post hoc Bonferroni.

CSHQ-DE Subscales	Pre-COVID-19	Post-COVID-19 Initial Presentation	Post-COVID-19 Re-Presentation	*p*-Value
Bedtime resistance	8.2 (± 2.7)	8.4 (± 2.4)	8.0 (± 2.7)	0.495
Sleep onset delay	1.8 (± 0.9)	2.0 (± 0.8)	2.0 (± 0.9)	0.239
Sleep duration	4.1 (± 1.3)	4.6 (± 1.4)	4.5 (± 1.6)	0.150
Sleep anxiety	5.4 (± 2.1)	5.5 (± 1.9)	5.3 (± 2.2)	0.665
Night waking	4.0 (± 1.6)	4.5 (± 1.6)	4.2 (± 1.5)	0.130
Parasomnia	9.0 (± 2.6)	9.3 (± 2.4)	9.1 (± 2.4)	0.645
Sleep-disordered breathing	3.5 (± 1.1)	3.5 (± 1.0)	3.6 (± 1.3)	0.543
Daytime sleepiness	12.6 (± 3.0)	14.4 (± 3.2)	13.2 (± 3.2)	< 0.001

**Table 6 children-09-01419-t006:** Treatment recommendations for non-organic sleep disorders.

Regular bedtime as part of a regular daily routine, including weekends (or put to bed only when the child is tired). Regular daytime sleep at the same time (particularly important for smaller children).
In the evening, avoid bright light, especially in the blue-wave range (LCD screens, mobile phones, tablets, TV, etc.). Avoidance of media consumption in the evening, especially in the hour before going to bed.
Sufficient physical activity during the day.
No exciting or strenuous activities before falling asleep.
Quiet, dark and warm sleeping environment. Sleep environment not associated with other activities (gaming, cell phones, television, homework, punishment, etc.).
Bedtime rituals depending on age (between 15 and 30 min).
Eating/drinking: ○ The introduction of a behavioral pattern in early childhood of being fed during the day and sleeping at night; ○ The avoidance of excessive amounts of stimulating drinks during the day; ○ The avoidance of stimulating drinks and sweets a few hours before bedtime; ○ Not responding to the intake wishes of a child, which only serve to delay sleep; ○ At bedtime, avoid hunger and thirst as well as large meals and a significant liquid intake.
Not responding to wishes/demands of the child that only serve to delay sleep.
For older children and adolescents, restriction of naps to 20 min and not after 3 p.m.

## Data Availability

IPD sharing plan: The study protocol and individual participant data underlying the results reported in the article after deidentification. Documents will be available 3 months after publication for 3 years for investigators submitting a structured application after approval by a committee. Data will be made available for all types of analyses that produce results in the approved application. Requests should be sent informally to daniel.vilser@med.uni-jena.de. A data agreement must be signed for access to the data.

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
