# Peer review of "The CSHQ-DE Questionnaire Uncovers Relevant Sleep Disorders in Children and Adolescents with Long COVID"

_children, 2022, doi:10.3390/children9091419_

Round 1
Reviewer 1 Report
Concerns about the impact of Covid-19 infection and sleep disturbance are relevant and timely in light of the current pandemic. This noncontrolled study demonstrates using a standardized children's sleep questionnaire. The paper is concise and well-written, and the results clearly demonstrate a link between Covid-19 infection and subsequent sleep problems in children. The results are clearly presented and it is assumed original findings. The authors present a solid discussion about the limitations of the study related to not having a control group. In addition, the authors might mention that potential other co-morbid sleep disorders such as OSA may present a confound. The mechanism for the sleep disturbance would add to the study. Otherwise, this is a significant study that is transparent and rigorous in light of the limitations.
Author Response
In addition, the authors might mention that potential other co-morbid sleep disorders such as OSA may present a confound
- This is an important point that we have now included in the discussion and limitations.
The mechanism for the sleep disturbance would add to the study
- The underlying mechanism for the sleep disorder remains unknown. We have discussed various theories in detail in chapter 4

Reviewer 2 Report
-
The topic is current.
2. The abstract is appropriate for both the title and the article.
3. The manuscript is well-organized.
4. This is a meaningful study as it contributes to the understanding of sleep disorders in children and adolescents with long COVID-19.
5. The statistical analysis is effectively presented in the article.
6. It is important to consider how the values of the survey can be expressed more clearly in the discussion section.
7. The conclusion could be strengthened by adding some objective statements about the contributions of this study.
.
Author Response
It is important to consider how the values of the survey can be expressed more clearly in the discussion section
- In the discussion section, we have sought to better classify the investigation results and implications for the future care of sick children.
The conclusion could be strengthened by adding some objective statements about the contributions of this study
- We have included the most important values in the conclusion. Two statements have been added to strengthen the conclusion.
